# The Tropomyosin Family as Novel Biomarkers in Relation to Poor Prognosis in Glioma

**DOI:** 10.3390/biology11081115

**Published:** 2022-07-26

**Authors:** Ke Huang, Huihui Wang, Jia Xu, Ruiming Xu, Zelin Liu, Yi Li, Zhaoqing Xu

**Affiliations:** 1Key Laboratory of Preclinical Study for New Drugs of Gansu Province, School of Basic Medical Sciences, Lanzhou University, Lanzhou 730030, China; huangk18@lzu.edu.cn; 2Key Laboratory of Dental Maxillofacial Reconstruction and Biological Intelligence Manufacturing, School of Stomatology, Lanzhou University, Lanzhou 730030, China; wanghhlzu@163.com (H.W.); xuj2021@lzu.edu.cn (J.X.); liuzl20@lzu.edu.cn (Z.L.); 3The Second Hospital of Dalian Medical University, Dalian 116027, China; xrm0920@outlook.com

**Keywords:** tropomyosin family, biomarker, prognosis, tumorigenicity, glioma

## Abstract

**Simple Summary:**

Due to the malignant features of glioma, current interventions result in limited treatment effects and poor prognoses for all patients. The functions of the tropomyosin (TPM) family in tumors and cancers have been explored. However, striking differences have been observed. This study aims to further our understanding of the effects of TPMs in glioma. Our study explored the expression and prognoses of TPM in glioma, as well as the gene functions of TPMs. High expression of TPM3 and TPM4 were positively correlated with poorer prognosis in glioma, and TPM3 could serve as a novel independent prognostic factor of glioma.

**Abstract:**

(1) Background: The functions of the tropomyosin (TPM) family in tumors and cancers have been explored; however, striking differences have been observed. This study aims to further our understanding of the effects of TPMs in glioma, and find novel biomarkers for glioma. (2) Methods: RNA-seq data were downloaded from TCGA and GTEx. Survival analyses, Cox regression, nomogram, calibration curves, ROC curves, gene function enrichment analyses, and immune cell infiltration analyses were carried out using R. CCK8 assay, while Brdu assay, colony formation assay, and Transwell assay were used to verify the functions of TPM3 in glioma. (3) Results: TPM1/3/4 were significantly more highly expressed in glioma than that in normal tissues, while higher expression of TPM2/3/4 was correlated with a worse overall survival than lower expression of TPM2/3/4. Furthermore, bioinformatic analyses indicated that TPM3/4 could be promoting factors for poorer survival in glioma, but only TPM3 could serve as an independent prognostic factor. Gene function analyses showed that TPMs may be involved in immune responses. Moreover, further experimental investigations verified that TPM3 overexpression enhanced the proliferation and tumorigenicity of glioma. (4) Conclusions: High expression of TPM3/4 was positively correlated with poorer prognosis in glioma, and TPM3 could serve as a novel independent prognostic factor of glioma.

## 1. Introduction

Glioma is one of the most common human central nervous system (CNS) tumors in human beings [1]. Currently, there are a variety of frontline interventions available for treating malignant gliomas, including surgery, chemotherapy, and radiotherapy. However, the highly infiltrative and invasive nature of glioma cells result in limited therapeutic effect and poor prognosis for all patients [2,3]. To effectively treat glioma, there is an urgent need to identify a potential biomarker that can serve both as a diagnostic and prognostic indicator, so that patients can benefit from more effective therapies and increased survival chances.

The tropomyosin (TPM) family, which is a group of actin-associated proteins, plays a major role in regulating the actin cytoskeleton. There are four TPM genes, TPM1, TPM2, TPM3, and TPM4, which can produce more than 40 different isoforms of TPM through an alternative exon-splicing mechanism [4,5]. As early as 1946, Bailey identified TPM in striated muscle, and its role in muscle contraction has been well characterized ever since [6,7]. Recently, many researchers have been focusing on TPM functions in tumors and cancers; however, striking differences have been observed. In breast tumors, oral squamous cell carcinoma (OSCC), and renal cell carcinoma (RCC), TPMs are suggested to be tumor suppressor genes [8,9,10]. In pancreatic cancer, TPM4 is thought to serve as a prognostic biomarker for pancreatic cancer [11]. In parallel, Helfman et al. suggested that TPMs are targets of oncogenic signaling and can function as regulators of oncogenic signaling [12]. Clearly, more studies are required to further understand the roles of the TPM family in human cancers.

Previous studies have characterized the role of TPMs in some cancers and tumors, but the importance of the type of TPM for prognostic biomarkers for glioma remains unclear. With the rapid development of gene sequencing technology and the establishment of various databases, comprehensive analysis of TPMs by bioinformatic analysis has become possible. In this study, we conducted an in-depth and comprehensive analysis of the expression of TPMs in glioma and evaluated their potential as prognostic biomarkers based on TCGA data, thus providing additional data to help clinicians more accurately assess the long-term outcome prognosis in patients with glioma.

## 2. Materials and Methods

### 2.1. Data Collection and Integration

The dataset consisted of 1846 samples (1152 normal samples from GTEx, 5 peritumor tissues from TCGA [13], and 689 tumor tissues from TCGA). The RNA-seq data and corresponding clinical information were downloaded from TCGA, and RNA-seq data in TPM format (from TCGA and GTEx), standardized by the Toil process, were downloaded from UCSC XENA (https://xenabrowser.net/datapages/) (accessed on 3 March 2022) [14]. Thereafter, log2-fold change (log2FC) was calculated for further comparison of the mRNA expression level between tumor and normal samples. Clinical information on the glioma patients included age, gender, WHO grade, IDH status, 1p/19q status, histological type, and overall survival (OS). Samples with unclear or incorrect information were excluded to avoid unreliable results. The DNA methylation data (in TPM format) were acquired from TCGA, as previously described.

### 2.2. Survival and Statistical Analyses

According to the median expression level of TPMs, patients were split into high and low expression groups. The association between TPM expression level and overall survival was assessed by Kaplan–Meier (KM) survival analyses using the R software (version 3.6.3, R Core Team, Vienna, Austria) and the R package (survminer, version 0.4.9 and survival, version 3.2.10) (https://CRAN.R-project.org/package=survminer) (http://cran.r-project.org/package=survival) (accessed on 3 March 2022).

### 2.3. Univariate and Multivariate Cox Regression Analyses

In order to ascertain whether TPM expression, gender, age, race, WHO grade, IDH status, 1p/19q status, and histological type were independent prognostic factors for survival of glioma patients, univariate and multivariate Cox regression were performed. Hazard ratios (HR) and 95% confidence intervals (CI) were calculated in this study, and the significance threshold was set at *p* < 0.05. R package (survival, version 3.2.10) was used for data processing.

### 2.4. Construction of Nomograms, Calibration Plots, and ROC Curves

The nomogram was used to predict cancer prognosis. The calibration curves were plotted to visualize the deviation of predicted probabilities. Time-dependent receiver operating characteristic (ROC) curves were generated for diagnostic analyses using R package (pROC, version 1.17.0.1 and ggplot2, version 3.3.3) (https://cran.r-project.org/web/packages/pROC/index.html) (https://ggplot2.tidyverse.org/) (accessed on 3 March 2022).

### 2.5. TPM-Related Gene Function Enrichment Analyses

Differentially Expressed Genes (DEG) were analyzed using R package (DESeq2, version 1.26.0) [15]. The threshold of log_2_FC > 2 and adjusted *p*-value < 0.05 were chosen to consider genes as differentially expressed. Gene Ontology (GO) and Kyoto Encyclopedia of Genes and Genomes (KEGG) analyses were performed to evaluate potential gene functions associated with TPMs based on the TCGA database with R package (org.Hs.eg.db, version 3.10.0 and clusterProfiler, version 3.14.3) [16]. Gene Set Enrichment Analysis (GSEA), a computational method that determines the statistical significance of a priori defined set of genes and the existence of concordant differences between two biological states, was performed using R package (clusterProfiler, version 3.14.3) [17] and R package (ggplot2, version 3.3.3) was also used for data visualization. In GSEA analyses, gene sets were evaluated by the absolute value of the Normalized Enrichment Score (NES), adjusted *p* value (p.adj), and false discovery rate (FDR). Gene sets with |NES| > 1, p.adj < 0.05 and FDR < 0.25 were considered to be significantly enriched. Protein–protein interaction (PPI) network analysis of ten interacting proteins correlated with TGIF1 were collected from the STRING database (https://cn.string-db.org/) (accessed on 3 March 2022) [18].

### 2.6. Immune Cell Infiltration Analyses

The Tumor Immune Estimation Resource (TIMER) (http://timer.cistrome.org/) (accessed on 3 March 2022) is a public database that aims at estimating the relative abundance of tumor-infiltrating immune cells. The correlation between TGIF1 expression and infiltrated immune cells was analyzed using TIMER 2.0 and R package (GSVA package, version 1.34.0) [19]. In this study, six types of infiltrating immune cells in low-grade glioma (LGG) and glioblastomas (GBM) were investigated in the TIMER database [20], and 24 types of immune cells in glioma were explored by R [21].

### 2.7. Cell Cultures

The human glioma U87-MG cells and U251 cells were purchased from the Cell Bank of the Chinese Academy of Sciences (Shanghai, China) and cultured in high glucose Dulbecco’s Modified Eagle Medium (DMEM) (BasalMedia, Shanghai, China), supplemented with 10% fetal bovine serum (FBS) (ABW, Shanghai, China), penicillin (100 U/mL) (BasalMedia), and streptomycin (100 U/mL) (BasalMedia). Cells were cultured in a humidified incubator and maintained at 37 °C and 5% CO_2_.

### 2.8. TPM3 Knockdown and TPM3 Overexpression

For TPM3 knockdown, the TPM3 and negative control (NC) siRNA were purchased from GenePharma (Shanghai, China). Cells were transfected using Lipofectamine™ 2000 (Invitrogen, Carlsbad, CA, USA) according to the manufacturer’s protocol. Briefly, 1 × 10^6^ cells were seeded in a T-25 cell culture flask in DMEM supplemented with 10% FBS. After incubation for 12 h, adherent cells were washed with PBS (Solarbio, Beijing, China) and then added to 2.0 mL serum-free Opti-MEM (Invitrogen) containing 30 nM of either NC or TPM3 siRNA and 30 nM Lipofectamine™ 2000. After transfection for 6 h, the culture medium was changed to complete medium. Eventually, two groups were set up: the RNAi_NC group (transfected with NC siRNA) and the RNAi_TPM3 group (transfected with TPM3 siRNA).

For TPM3 overexpression, the lentiviral was purchased from GenePharma. Cells were plated at a concentration of 5 × 10^4^ cells per well in a 6-well plate. After incubation for 12 h, adherent cells were washed with PBS and then added to 3.0 mL serum-free medium containing 8 μg/mL polybrene (Solarbio, Beijing, China). Finally, cells were infected for 24 h with lentivirus at a multiplicity of infection (MOI) of 50. For screening of stable TPM3-overexpressed cell lines, cells were selected with 2.5 μg/mL puromycin (Solarbio, Beijing, China) for 3 days. Eventually, two groups were set up: the OE_NC group (transfected with the empty vectors) and the OE_TPM3 group (transfected with the TPM3 lentiviral vector).

Gene knockdown or overexpression efficiency was determined by western blot.

### 2.9. Western Blot

Proteins were extracted from cells using RIPA Lysis Buffer (Solarbio, Beijing, China) containing 1.0% PMSF (Solarbio, Beijing, China). Protein concentrations were quantified using the BCA Protein Assay Kit (Solarbio, Beijing, China). Next, equal amounts of protein samples were loaded and separated on SDS-PAGE gels (Solarbio, Beijing, China). The blots of gels were transferred onto PVDF membranes (Millipore, Burlington, MA, USA). The membranes were blocked with 5% non-fat milk (BD, NJ, USA) for 1.5 h at room temperature, then incubated with primary antibodies overnight at 4 °C, followed by the secondary antibody at room temperature for 1 h. The chemiluminescent signals of each protein band were processed with enhanced chemiluminescence (ECL) reagent (Yeasen Biotechnology, Shanghai, China) and detected by ChemiScope Capture (Clinx Science Instruments, Shanghai, China). The primary antibodies were TPM3 (Millipore, Burlington, MA, USA) and GAPDH (Proteintech, Wuhan, China) at a ratio of 1:1000.

### 2.10. Cell Growth and Cell Proliferation Analyses

Cell growth curves were assessed by using the Cell Counting Kit-8 (CCK-8) (Abmole, Houston, TX, USA). Cells were seeded at a density of 2000 cells per well in 96-well plates and incubated for 1, 2, and 3 days, respectively. Then, the cells were incubated with CCK-8 solution for 2 h. The optical density (OD) values were measured on the microplate reader (Infinite M200 Pro, Tecan Group, Männedorf, Switzerland) at a wavelength of 450 nm. Similarly, cell proliferation was quantified on day 2 using a 5-bromo-2-deoxyuridine (Brdu) Cell Proliferation Assay Kit (BioVision, Milpitas, CA, USA).

### 2.11. Cellular Migration Assays

The migration ability of cells was assessed using Transwell chambers (8.0 μm pore size, Corning, CA, USA). Cells were resuspended with serum-free medium at a concentration of 5 × 10^5^ cells/mL and 200 μL cell suspensions were plated into the upper Transwell chambers. Next, 700 μL DMEM containing 10% FBS was added to the lower compartments. Following 24 h of incubation, the cells were fixed with 4% paraformaldehyde and stained with 0.1% crystal violet. The cells in the upper chambers were removed using cotton buds. Eventually, the visible cells which had migrated through the membranes were photographed by microscope, and the results were quantified using ImageJ software.

### 2.12. Statistical Analyses

The statistical calculations and graphing were processed through the R software (version 3.6.3, R Core Team, Vienna, Austria) and Adobe Illustrator software (version 25.0.0.60, Adobe Inc., Mountain View, CA, USA). The correlations between clinical information and gene expression were assessed using Cox regression. *p* < 0.05 was the cut-off criterion.

## 3. Results

### 3.1. Clinical Characteristics of Glioma

The data of 698 primary tumors and 5 peritumor tissues were downloaded from the TCGA database, and the data of 1152 normal samples were downloaded from GTEx. Clinical information included age, gender, WHO grade, IDH status, 1q/19p codeletion, primary therapy outcome, histological type, and OS event. Supplementary data of WHO grade, IDH status, and 1q/19p codeletion were from a study by Ceccarelli M et al. [22]. Samples with unclear or incorrect information were excluded (Table 1).

### 3.2. TPMs Expression in Glioma Patients

The gene expression level was analyzed based on data from TCGA. The results showed that TPM1, TPM3, and TPM4 were significantly more highly expressed in glioma in comparison to normal tissues (Figure 1A), while TPM2 mRNA expression showed no statistical differences. Next, KM survival analysis was performed to ascertain the association between TPM expression and overall survival of glioma patients. The KM curves showed that a high expression level of TPM2, TPM3, and TPM4 was significantly correlated with worse overall survival (Figure 1B–E), while no significant difference was observed in KM analysis for TPM1. The results indicated that TPMs, especially TPM3 and TPM4, may function as oncogenes, and a high expression of TPM3/4 could be correlated to a worse prognosis outcome in glioma.

### 3.3. Correlations between Clinical Characteristics and TPM Expression of Glioma

Correlations between clinical characteristics and TPM mRNA expression in glioma were analyzed by the Kruskal–Wallis test. The results showed that TPM2, TPM3, and TPM4 were more expressed in high-grade gliomas than in low-grade gliomas, while for TPM1, the trend was not as evident as in other groups (Figure 2A). Investigations of the relationship between TPM expressions and IDH status showed that a lower expression level of TPM2, TPM3, or TPM4 was correlated with IDH mutation, while TPM1 showed no obvious difference (Figure 2B). Next, the relationships between TPM expressions and 1p/19q status were evaluated, and only TPM3 and TPM4 were found to be associated with 1p/19q non-codeletion (Figure 2C). Correspondingly, higher expressions of TPM2, TPM3, and TPM4 were associated with older age groups (Figure 2D). As for gender, only TPM1 expression was found to be slightly higher in males than females (Figure 2E). Of note, only TPM3 and TPM4 were found to be correlated with primary therapy outcomes, and a higher expression of TPM3 or TPM4 was correlated to a worse primary therapy outcome (Figure 2F).

### 3.4. Correlations between Clinical Characteristics and Prognosis Associated with TPM3 and TPM4

Given that the association between TPM3/TPM4 expression and the clinical characteristics was so striking, we further explored the prognosis associated with TPM3 and TPM4 in different clinical contexts. In WHO grade III and IV, high expressions of both TPM3 and TPM4 were significantly correlated with a poorer prognosis, while the trend was not as evident in grade II (Figure 3A,B and Figure 4A,B). In parallel, high expressions of TPM3 and TPM4 were markedly associated with a worse prognosis in mutant IDH status and 1p/19q non-codeletion status (Figure 3C–F and Figure 4C–F). High expression of TPM3 and TPM4 was found to be related to a worse prognosis in all age groups (Figure 3G,H and Figure 4G,H). With regard to the histological types of glioma, high expressions of TPM3 and TPM4 were evidently associated with poorer prognosis in astrocytoma and oligodendroglioma (Figure 3I,K and Figure 4I,K). No significant relationship was found between TPM3/4 expression and prognosis of glioblastoma (Figure 3J and Figure 4J).

### 3.5. Diagnostic Value of TPMs in Glioma

Univariate and multivariate Cox regression analyses were used to ascertain the independent risk factors. Interestingly, the univariate Cox regression analyses indicated that a high expression of TPM2/3/4 was a promoting factor for poorer survival of glioma, while the multivariate Cox regression analyses indicated that a high expression of TPM3 was an independent prognosis factor for poor prognosis (Figure 5). In parallel, age, WHO grade, IDH status, 1p/19q status, and histological type served as risk factors as well (Figure 5). Next, we constructed nomograms with these prognosis factors to predict the 1-, 3- and 5-year survival probability (Figure 6A). The calibration plots for the nomogram-predicated survival probability were very close to the ideal reference line (Figure 6B). In addition, to better quantify the accuracy of the prediction of the prognostic effect of TPMs, the diagnostic values of TPM mRNA expressions were evaluated by ROC curves. The results showed that the Area Under the Curve (AUC) of TPM1 was 0.863 (Figure 6C), the AUC of TPM2 was 0.480 (Figure 6D), the AUC of TPM3 was 0.721 (Figure 6E), and the AUC of TPM4 was 0.649 (Figure 6F). Since excellent diagnostic effects of TPM3 and TPM4 were recognized, we further conducted several ROC analyses combining TPM3/TPM4 mRNA expression with some key clinical characteristics (WHO grade, IDH status, and 1p/19q status) (Figure 7). The results above indicated that TPM3 and TPM4 could serve as disadvantageous factors for the survival of glioma, while TPM3 served as an independent predictor of a bad prognosis.

### 3.6. Predicted Gene Functions of TPM3 and TPM4

GO and KEGG analyses were performed to explore the potential biological functions related to TPM expression. The outcomes indicated that a series of functions are associated with both TPM3 and TPM4 expression, such as cytokine–cytokine receptor interaction, immunoglobulin receptor binding, immunoglobulin complex, humoral immune response mediated by circulating immunoglobulin, and complement activation (Figure 8). A range of pathways linked to both TPM3 and TPM4 that met the significance threshold (|NES| > 1, p.adj < 0.05 and FDR < 0.25) were shown by GSEA analyses. Three key pathways were presented in our study, which included immunoregulatory interactions between a lymphoid and a non-lymphoid cell, interactions between immune cells and microRNAs in the tumor microenvironment, and cancer immunotherapy by PD1 blockade (Figure 9). In addition, genes correlated with TPMs were obtained and the PPI network analysis was mapped as shown in Figure 10. The abovementioned results indicated the potential role of TPMs in the development of glioma.

### 3.7. The Correlation between TPM3/TPM4 and Immune Cell Infiltration in Gliomas

Given that a close connection was found between the immunomodulatory signaling pathways and TPM mRNA expression, we further explored the relationships between TPM3/TPM4 and immune cell infiltration. To begin, we investigated the six representative types of infiltrating immune cells in LGG and GBM, respectively, using the TIMER database. TPM3 and TPM4 expression showed positive correlations with the immune cell infiltrations in LGG, while the trend in GBM was not as evident (Figure 11A). Moreover, we explored 24 types of immune cells using data downloaded from TCGA. The results indicated that the expression level of TPM3 and TPM4 had obvious positive correlations with infiltrating levels of macrophages, Th2 cells, neutrophils, eosinophils, activated DC (aDCs), immature DCs (iDCs), T cells, NK cells, NK CD56dim cells, cytotoxic cells, and B cells (Figure 11B,C).

### 3.8. Overexpression of TPM3 Enhanced the Proliferation and Migration of Glioma Cells

Previous analyses indicated that TPM3 is the most representative oncogene for the TPM family in glioma. Therefore, to ensure the correlation between TPM3 and several representative hallmarks of glioma [23,24], TPM3 was overexpressed in U87-MG and U251 cells using lentivirus, and Western blot was conducted to confirm the TPM3 overexpression (Figure 12A). Next, the growth curves of glioma showed that U87 cells and U251 cells grew faster when TPM3 was overexpressed (Figure 12B). The results of Brdu assay determined the stimulatory effect of TPM3 overexpression on U87 cell proliferation (Figure 12C). Subsequently, the results of the Transwell migration assay indicated that the migration ability increased when TPM3 was overexpressed (Figure 12D,E).

### 3.9. Knockdown of TPM3 Impaired the Proliferation and Migration of Glioma Cells

To further verify the biological effects of TPM3, RNAi technology was used for TPM3 knockdown, and Western blotting was conducted to confirm the TPM3 knockdown efficiency (Figure 12A). The growth curves showed that the growth rates of both U87 and U251 cells slowed down when TPM3 was knocked down (Figure 12B). The results of Brdu assay corroborated the results of CCK8. In addition, the Transwell migration assay was employed and the results indicated that the migration ability decreased when TPM3 was knocked down (Figure 12D,E).

These results were consistent with the bioinformatic analyses and indicated that TPM3 may enhance the proliferation and tumorigenicity of glioma cells.

## 4. Discussion

Glioma is one of the most common human CNS tumors, and current front-line interventions available for treating malignant gliomas may not be beneficial to all patients. To address this concern, a potential biomarker and therapeutic target that can serve both as a diagnostic and prognostic indicator is urgently required.

In this study, it was found that TPM1, TPM3, and TPM4 were significantly upregulated in gliomas, while a high expression level of TPM2, TPM3, and TPM4 was significantly correlated with poorer prognosis. Next, correlations between clinical characteristics and TPM mRNA expression in glioma were analyzed. Mutations in IDH have been reported in a variety of cancers [25]. In gliomas, IDH mutations were initially identified in a high percentage of LGGs [26], and were widely considered to be associated with a better prognosis [27,28,29]. Chromosome 1p/19q co-deletion (codel) has been recognized as a diagnostic and prognostic marker since 1998 [30]. The results showed that a high expression of TPM2, TPM3, and TPM4 was significantly correlated with age, WHO grade, IDH status, and 1p/19q status, and the association between TPM3/TPM4 expression and the clinical characteristics was striking. Further subgroup analyses for TPM3 and TPM4 showed that a high expression of TPM3/TPM4 was significantly correlated with poorer prognosis in various clinical contexts. In addition, the results of univariate and multivariate Cox regression analyses, nomograms, calibration plots, and ROC curves, taken together, indicated that TPM3 and TPM4 could serve as predictors of a poor prognosis, while TPM3 served as an independent predictor of a bad prognosis.

The functions of the TPM family in tumors and cancers have been explored extensively; however, striking differences have been observed [12] and the role of the TPM family in glioma remains unknown. Consequently, we explored the gene functions associated with TPM3 and TPM4 in glioma. The outcomes indicated that the potential biological functions may involve cytokine–cytokine receptor interaction, immunoglobulin receptor binding, immunoglobulin complex, humoral immune response mediated by circulating immunoglobulin, and complement activation.

The tumor microenvironment (TME) plays an important role in the immunosuppressive feature in glioma [31,32,33,34], and immune infiltrate changes at each tumor stage and specific group of cells have a major impact on survival [21]. Active communication among tumor cells, neighboring healthy cells, and the adjacent immune environment promotes the cancerogenic processes and drives therapy resistance [35]. Immune cells are important constituents of the tumor stroma and the infiltration of immune cells varies at each tumor stage [21]. Growing evidence suggests that the innate immune cells (macrophages, neutrophils, dendritic cells, innate lymphoid cells, myeloid-derived suppressor cells, and natural killer cells), as well as adaptive immune cells (T cells and B cells), contribute to tumor progression when present in the TME [21]. In this study, our findings showed that TPM3 and TPM4 expression were strongly positively correlated with infiltrating levels of macrophages, Th2 cells, Th17 cells, neutrophils, eosinophils, aDCs, iDCs, T cells, NK cells, NK CD56dim cells, cytotoxic cells, and B cells, and further aggravated immunosuppression. DCs are known to initiate pathogen-specific T cell responses and are therefore important for bolstering protective immunity. During the adaptive immune response process, DCs recognize, capture, and present antigens, upregulate costimulatory molecules, produce inflammatory cytokines, and then travel to secondary lymphoid organs for antigen presentation to T cells. Analogous to tumor-associated macrophages (TAMs), DCs can be stratified into specific subtypes. Th2 and Th17 infiltration were expected to contribute to the immune suppression type of DCs, as shown in previous studies [36]. Infiltration of neutrophils was also observed in our analysis. Neutrophils account for up to 70% of circulating leukocytes and are the first line of defense against pathogens [37]. In the context of cancer, the phenotype of neutrophils depends on tumor type and the stage of tumor progression. As the tumor progresses, they adopt an immunosuppressive phenotype. TPM3 and TPM4 showed an intimate connection with immune cell infiltration, and further research in this area is required.

Since previous bioinformatic analyses indicated that TPM3 is the most representative oncogene for the TPM family in glioma, TPM3 was overexpressed or knocked down in U87-MG cells and U251 cells to further investigate the biological functions in glioma. CCK8 and Brdu assays both indicated the stimulatory effect of TPM3 overexpression on U87 cell proliferation. In addition, Transwell migration assay indicated that the migration ability increased when TPM3 was overexpressed. TPM3 overexpression enhanced the proliferation and tumorigenicity of glioma cells.

## 5. Conclusions

Taken together, our preliminary findings revealed that a high expression of TPM3 and TPM4 was strongly and positively correlated with poorer prognosis in glioma, and TPM3 could serve as a novel independent prognostic factor in glioma.

## Figures and Tables

**Figure 1 biology-11-01115-f001:**
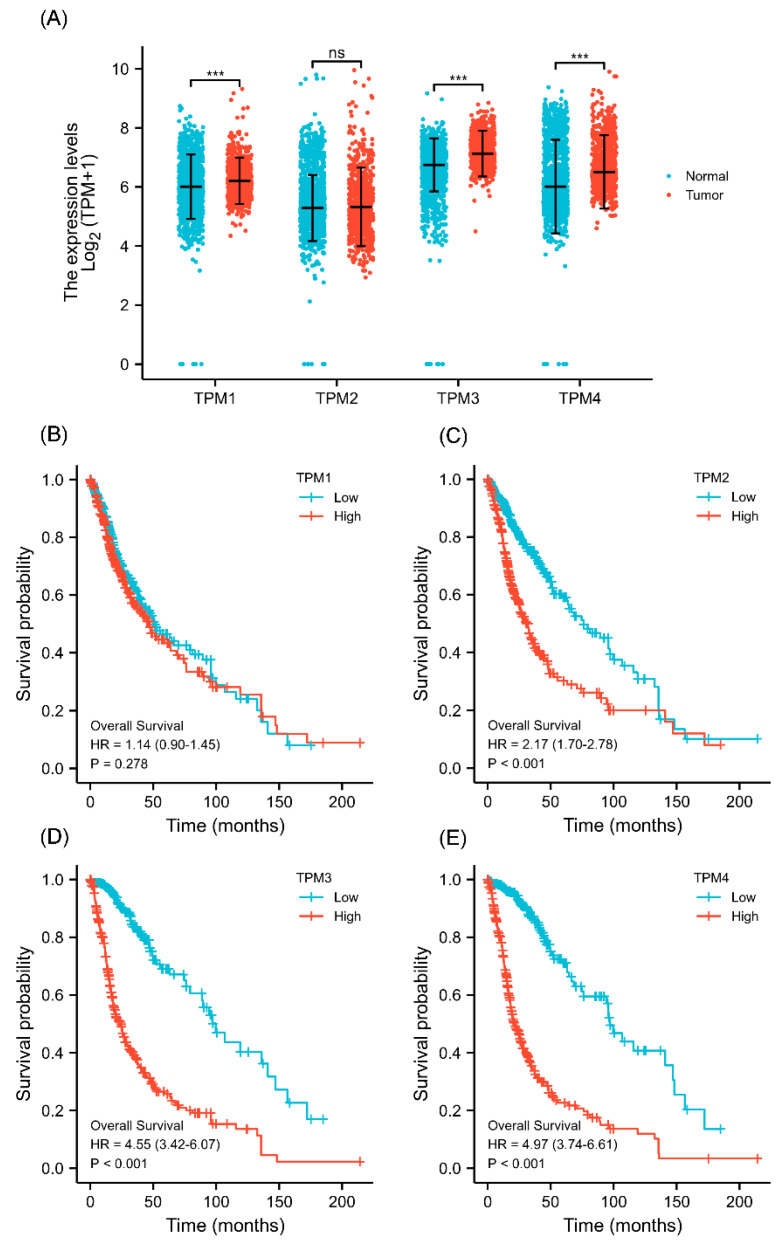
The TPM expressions and survival analyses in gliomas. (**A**) The differential expression analyses of TPMs in glioma patients. (**B**) KM curves of the associations between TPM1 expression and overall survival. (**C**) KM curves of the associations between TPM2 expression and overall survival. (**D**) KM curves of the associations between TPM3 expression and overall survival. (**E**) KM curves of the associations between TPM4 expression and overall survival. (ns, no significance; ***, *p* < 0.001).

**Figure 2 biology-11-01115-f002:**
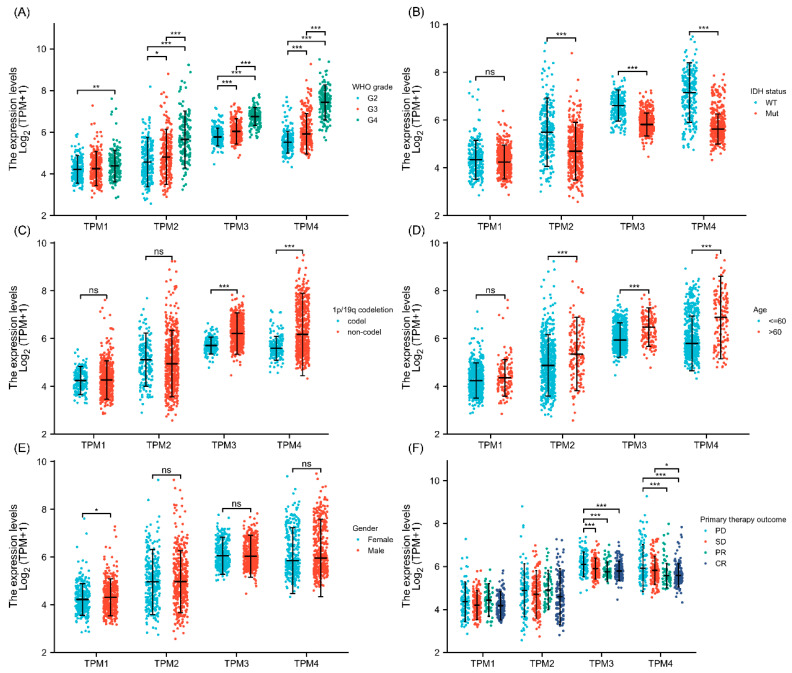
The association between TPM expressions and clinical characteristics. (**A**) Correlations between TPM expressions and WHO grade. (**B**) Correlations between TPM expressions and IDH status. (**C**) Correlations between TPM expressions and 1p/19q status. (**D**) Correlations between TPM expressions and age. (**E**) Correlations between TPM expressions and gender. (**F**) Correlations between TPM expressions and primary therapy outcomes (PD: progressive disease; SD: stable disease; PR: partial response; CR: complete response). (ns, no significance; *, *p* < 0.05; **, *p* < 0.01; ***, *p* < 0.001).

**Figure 3 biology-11-01115-f003:**
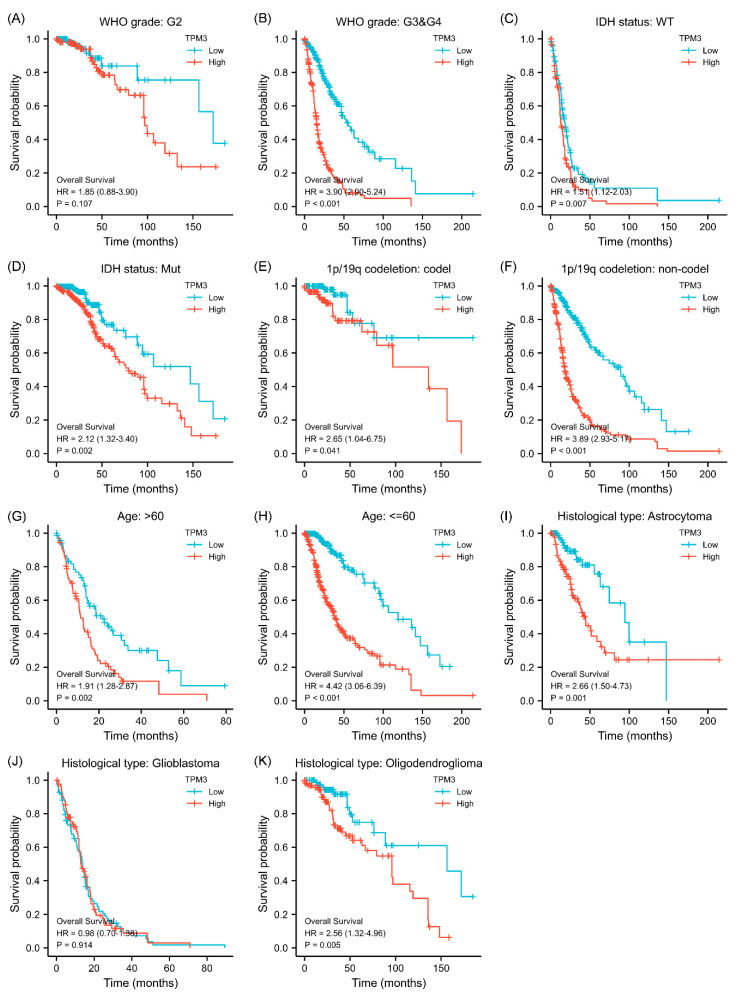
Subgroup survival analyses for the association between TPM3 expression and clinical characteristics. (**A**,**B**) Correlations between TPM3 expression and WHO grade. (**C**,**D**) Correlations between TPM3 expressions and IDH status. (**E**,**F**) Correlations between TPM3 expression and 1p/19q status. (**G**,**H**) Correlations between TPM3 expression and age. (**I**–**K**) Correlations between TPM3 expression and histological type.

**Figure 4 biology-11-01115-f004:**
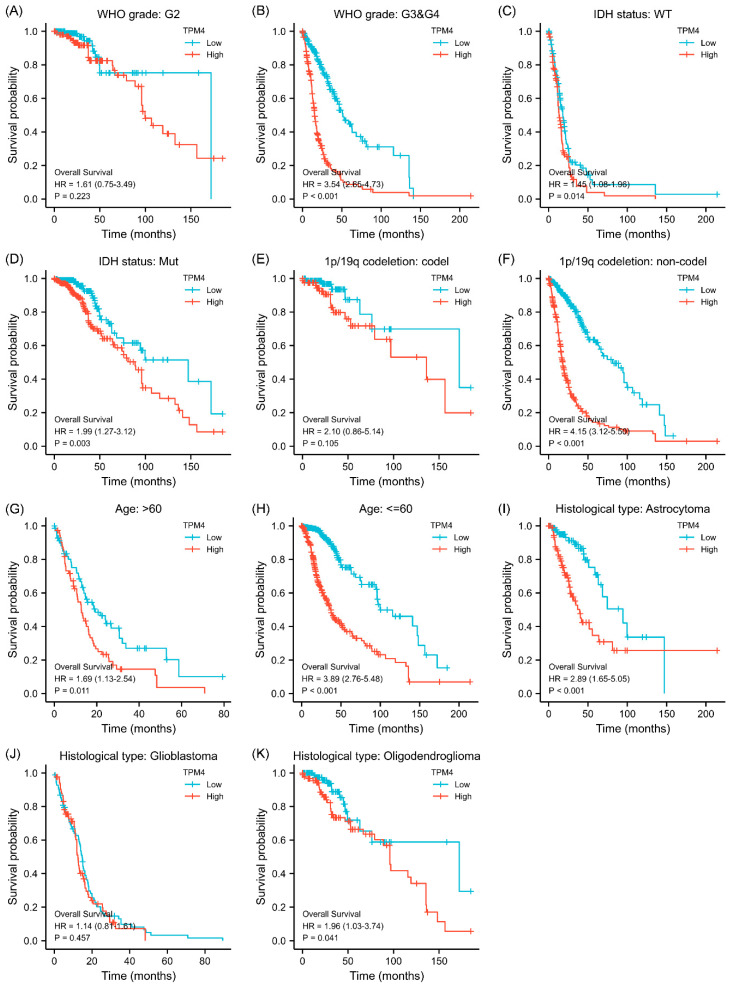
Subgroup survival analyses for the association between TPM4 expression and clinical characteristics. (**A**,**B**) Correlations between TPM4 expression and WHO grade. (**C**,**D**) Correlations between TPM4 expressions and IDH status. (**E**,**F**) Correlations between TPM4 expression and 1p/19q status. (**G**,**H**) Correlations between TPM4 expression and age. (**I**–**K**) Correlations between TPM4 expression and histological type.

**Figure 5 biology-11-01115-f005:**
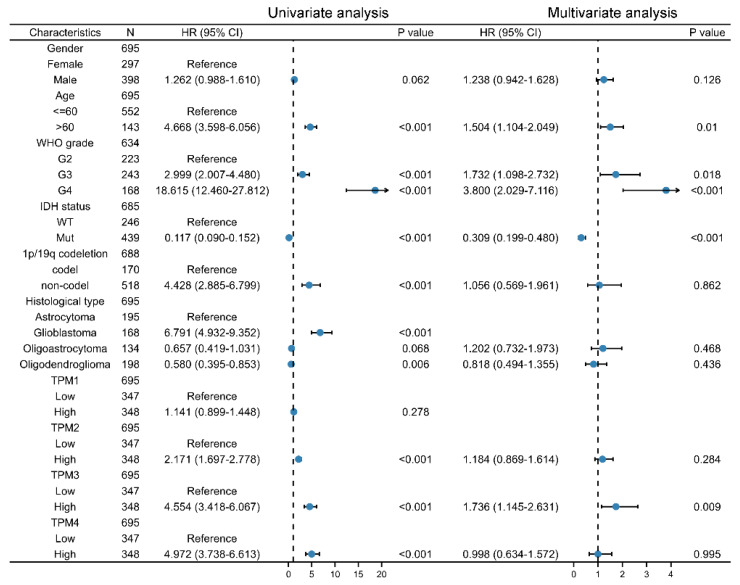
Univariate and multivariate Cox regression analysis risk score of TPM expressions and related key clinical characteristics. HR > 1 indicates disadvantageous factors, and HR < 1 indicates protective factors.

**Figure 6 biology-11-01115-f006:**
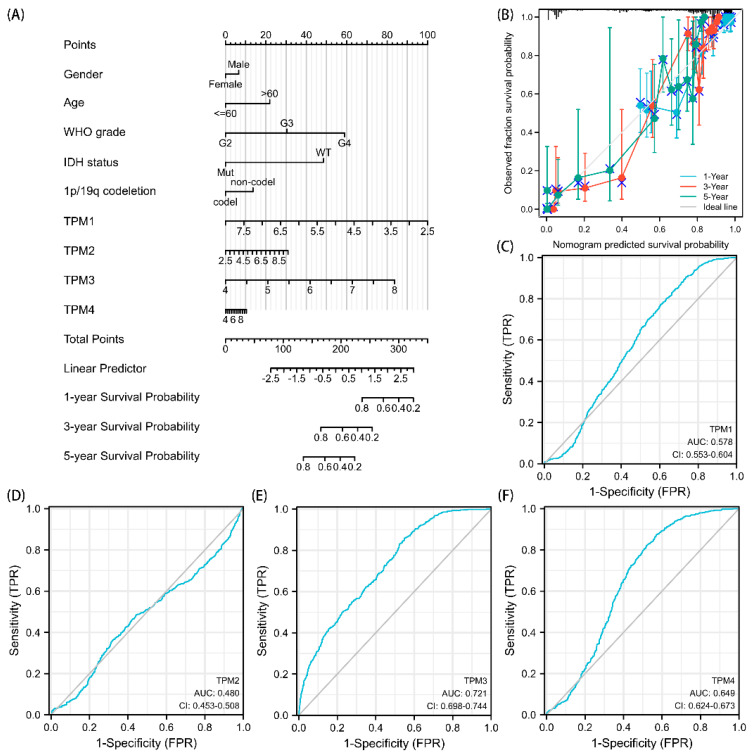
Diagnostic value of TPMs in glioma. (**A**) The nomogram was developed by integrating the TPM expressions with key clinical characteristics. (**B**) The calibration plot of the nomogram for predicting overall survival at 1 year, 3 years, and 5 years. (**C**) The diagnostic value of TPM1 mRNA expression was evaluated using a ROC curve. (**D**) The diagnostic value of TPM2. (**E**) The diagnostic value of TPM3. (**F**) The diagnostic value of TPM4.

**Figure 7 biology-11-01115-f007:**
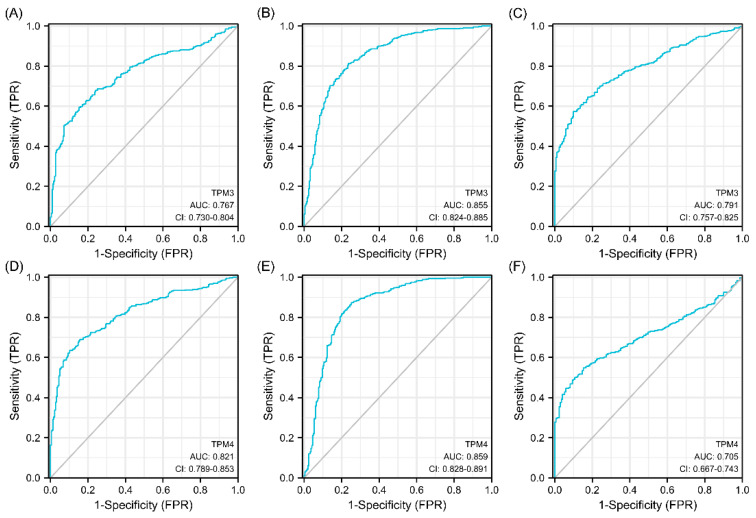
Diagnostic value of TPM3 and TPM4 in glioma. (**A**) The diagnostic value of TPM3 mRNA expression for the WHO grade. (**B**) The diagnostic value of TPM3 mRNA expression for the IDH status. (**C**) The diagnostic value of TPM3 mRNA expression for the 1p/19q status. (**D**) The diagnostic value of TPM4 mRNA expression for the WHO grade. (**E**) The diagnostic value of TPM4 mRNA expression for the IDH status. (**F**) The diagnostic value of TPM4 mRNA expression for the 1p/19q status.

**Figure 8 biology-11-01115-f008:**
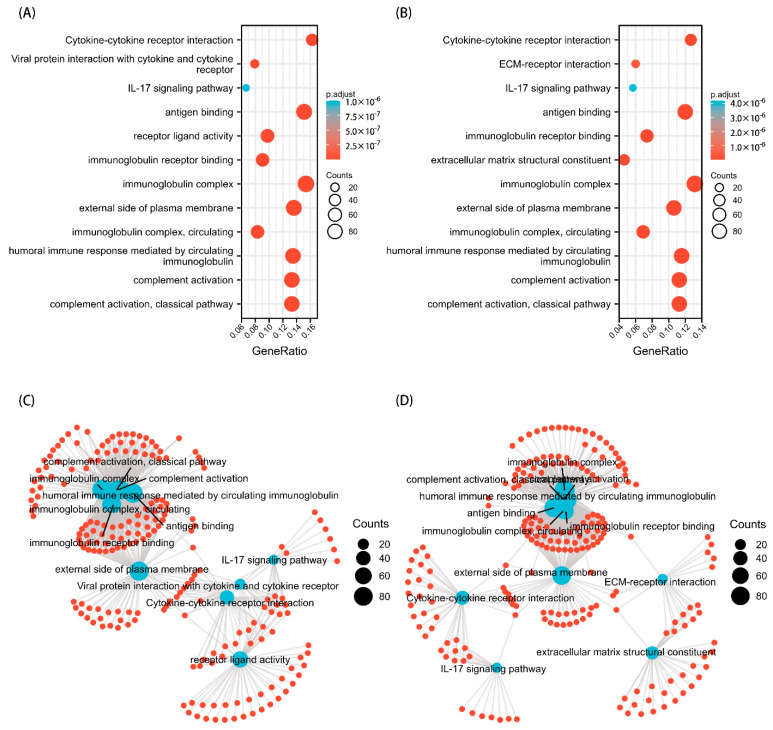
GO/KEGG enrichment analysis. (**A**) GO/KEGG analysis for TPM3. (**B**) GO/KEGG analysis for TPM4. (**C**) Network visualization of GO/KEGG enrichment analysis for TPM3. (**D**) Network visualization of GO/KEGG enrichment analysis for TPM4.

**Figure 9 biology-11-01115-f009:**
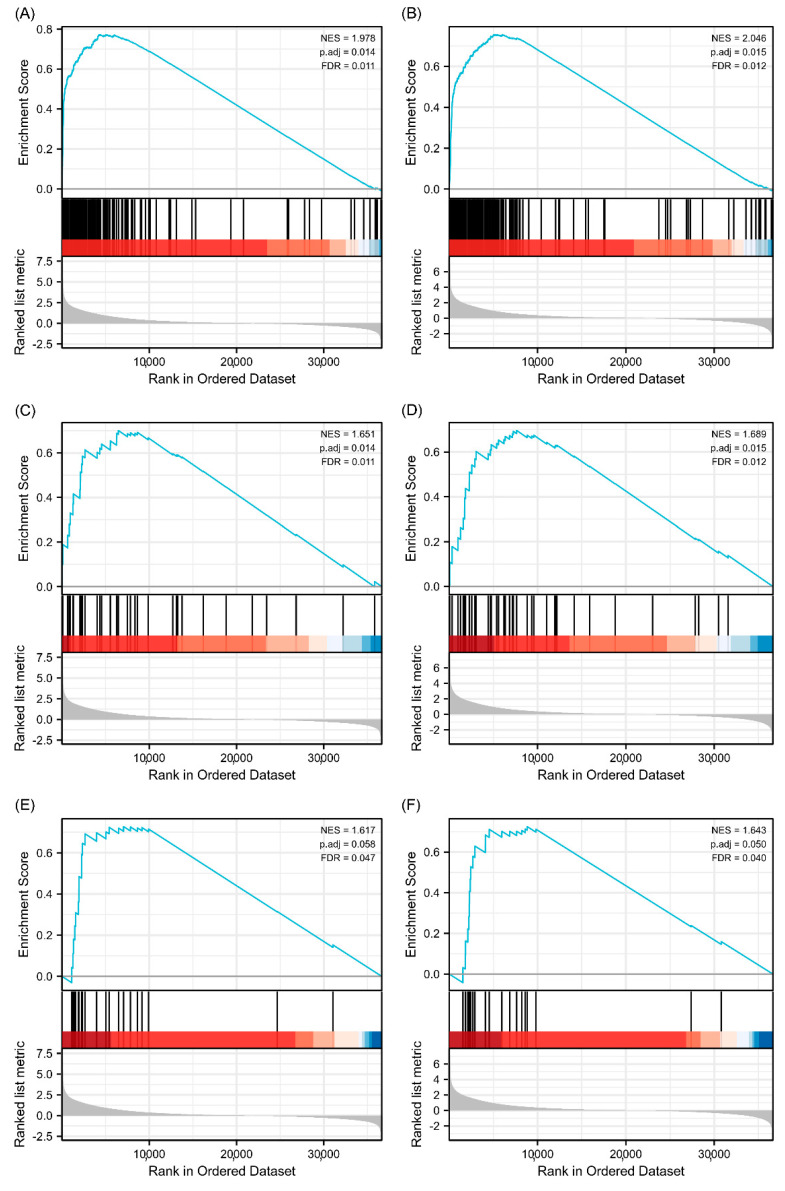
Enrichment plots from gene enrichment analysis (GSEA). (**A**) GSEA analysis for TPM3: immunoregulatory interactions between a lymphoid and a non-lymphoid cell. (**B**) GSEA analysis for TPM4: immunoregulatory interactions between a lymphoid and a non-lymphoid cell. (**C**) GSEA analysis for TPM3: interactions between immune cells and microRNAs in tumor microenvironment. (**D**) GSEA analysis for TPM4: interactions between immune cells and microRNAs in tumor microenvironment. (**E**) GSEA analysis for TPM3: cancer immunotherapy by PD1 blockade. (**F**) GSEA analysis for TPM4: cancer immunotherapy by PD1 blockade.

**Figure 10 biology-11-01115-f010:**
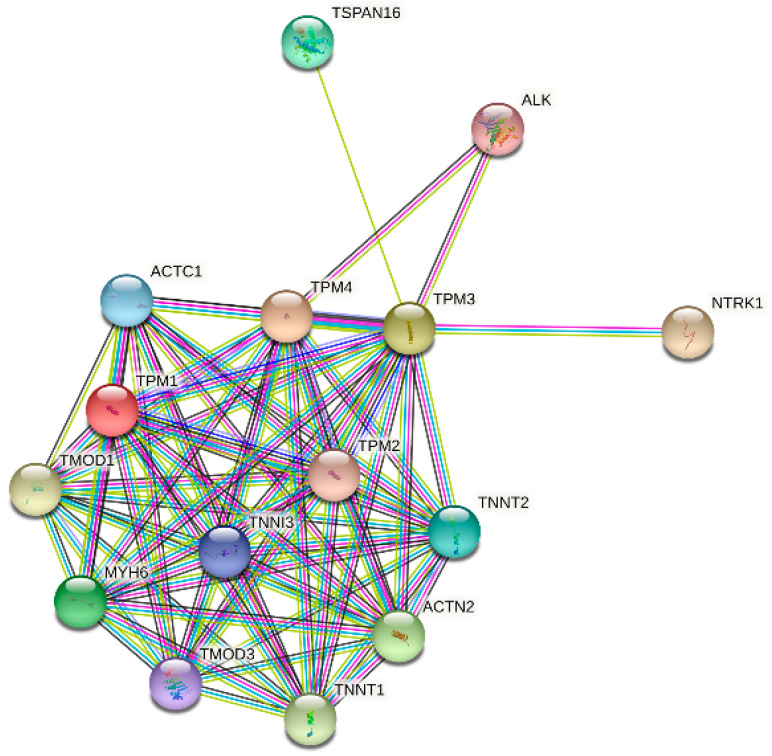
Protein–protein interaction (PPI) network analysis of 13 interacting proteins correlated with TPMs.

**Figure 11 biology-11-01115-f011:**
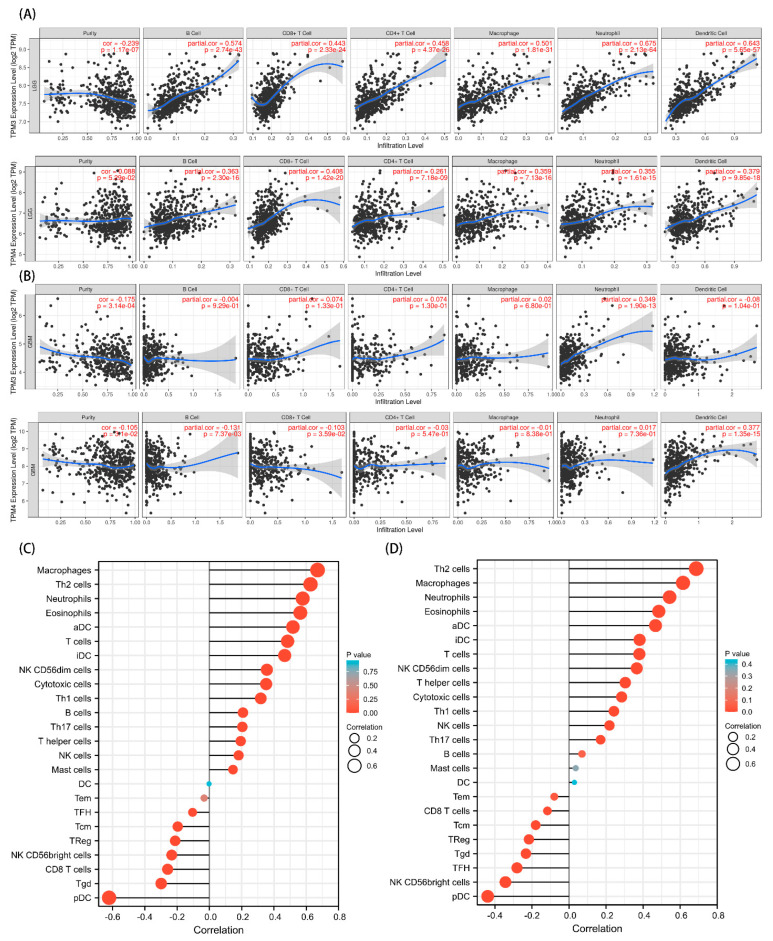
Correlation analysis between TPM3 and TPM4 expressions and immune cell filtration. (**A**) The correlation of TPM3/TPM4 with the six types of immune cell filtration levels in LGG based on the TIMER database. (**B**) The correlation of TPM3/TPM4 with the six types of immune cell filtration levels in GBM based on the TIMER database. (**C**) The correlation of TPM3 with 24 types of immune cells in glioma based on TCGA data. (**D**) The correlation of TPM4 with 24 types of immune cells in glioma based on TCGA data.

**Figure 12 biology-11-01115-f012:**
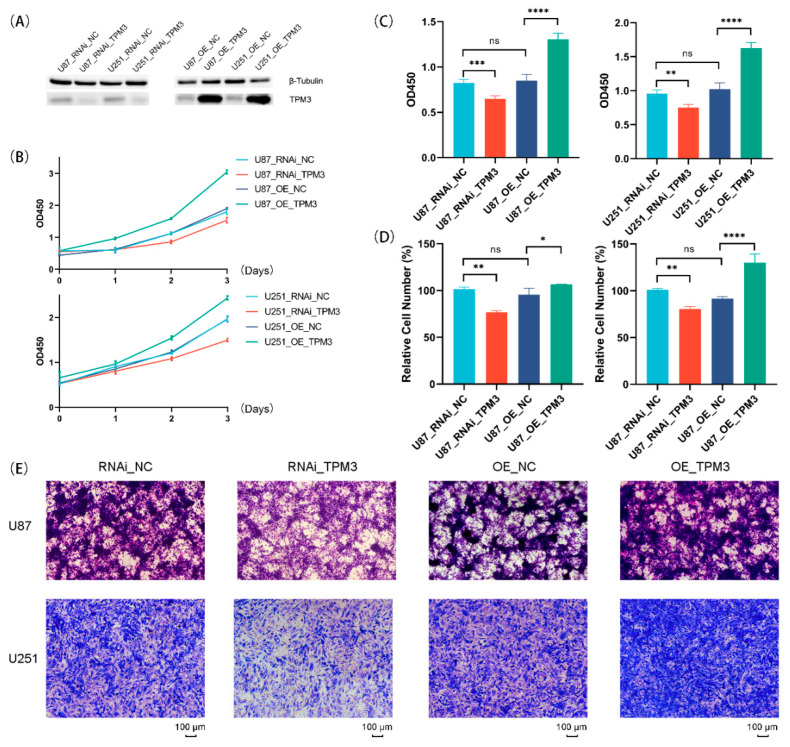
Correlations between TPM3 overexpression/knockdown and biological characterizations of U87 and U251 cells. (**A**) Western blot analyses were carried out to confirm the overexpression or knockdown efficiency of TPM3. (**B**) The growth curves of U87 cells and U251 cells with or without TPM3 overexpression/knockdown. (**C**) The proliferation ability of U87 cells and U251 cells quantified by Brdu assay. (**D**) Quantification of the Transwell assay. (**E**) Migration ability of U87 and U251 cells employed by Transwell migration assay. (ns, no significance; *, *p* < 0.05; **, *p* < 0.01; ***, *p* < 0.001; ****, *p* < 0.0001).

**Table 1 biology-11-01115-t001:** Clinical characteristics of the glioma patients.

Characteristic	Levels	Overall
n		696
Age, n (%)	≤60	553 (79.5%)
	>60	143 (20.5%)
Age, median (IQR)		45 (34, 59)
Gender, n (%)	Female	298 (42.8%)
	Male	398 (57.2%)
WHO grade, n (%)	G2	224 (35.3%)
	G3	243 (38.3%)
	G4	168 (26.5%)
IDH status, n (%)	WT	246 (35.9%)
	Mut	440 (64.1%)
1p/19q codeletion, n (%)	codel	171 (24.8%)
	non-codel	518 (75.2%)
Primary therapy outcome, n (%)	PD	112 (24.2%)
	SD	147 (31.8%)
	PR	64 (13.9%)
	CR	139 (30.1%)
Histological type, n (%)	Astrocytoma	195 (28%)
	Glioblastoma	168 (24.1%)
	Oligoastrocytoma	134 (19.3%)
	Oligodendroglioma	199 (28.6%)
OS event, n (%)	Alive	424 (60.9%)
	Dead	272 (39.1%)
DSS event, n (%)	Alive	431 (63.9%)
	Dead	244 (36.1%)
PFI event, n (%)	Alive	350 (50.3%)
	Dead	346 (49.7%)

## Data Availability

The datasets analyzed for this study are freely available in the TCGA (http://xena.ucsc.edu/) and UCSC XENA (https://xenabrowser.net/datapages/?cohort=GTEX&removeHub=https%3A%2F%2Fxena.treehouse.gi.ucsc.edu%3A443) (accessed on 3 March 2022).

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
