# Peer review of "The Tropomyosin Family as Novel Biomarkers in Relation to Poor Prognosis in Glioma"

_biology, 2022, doi:10.3390/biology11081115_

Round 1
Reviewer 1 Report
The aim of this study was to investigate the expression and role of tropomyosin (TPM) in gliomas. TPM1/3/4 were overexpressed in glioma. TPM2/3/4 were predictors of poor outcomes. Gene function analyses showed that TPMs may be involved in immune responses. TPM3 overexpression increased the proliferation and tumorigenicity of 31 gliomas. The authors concluded that TPM3 could serve as a novel independent prognostic factor of glioma.
This study could have been interesting but is hampered by methodological biaises. Moreover, the language has to be reviewed.
- The quality of the language is very poor. The manuscript should have been read by an English native speaker before submission!
- Unsuitable data are presented in the abstract. The authors mention that TPM1/3/4 were significantly higher expressed in glioma. Higher than what? They also state that TPM2/3/4 were correlated with a worse overall. Worse than what?
- In the introduction, from line 59 to 66, the authors expose methods and results. These data must be removed from the introduction. Conversely, after reading the introduction, the reader still not knows the aim of the study!
- Regarding the results based on the TCGA data, it is very strange to have chosen to mix all gliomas subtypes. The natural history and prognosis of gliomas is very heterogenous, varying widely between 1p19q codeleted low-grade and grade 4 IDH WT glioblastomas. Line 192-193, the authors conclude that high expression of TPMs could be predictors of a worse prognosis in glioma. I do not understand which data sustain this statement as the expression of TPM has not been studied according to glioma grade.
- In Figure 1, I guess that low/high level of TPM expression was defined by values inferior or superior to the median expression, respectively, but this data is unfortunately not precised.
- Mixing all glioma subtypes raises major problem also in Figure 2. Glioma grade and molecular profile are correlated to the age and the outcomes. Consequently, I do not think that analyses exposed in Figure 2 bring relevant data.
- In Figure 3, the analyses conducted based on the presence or not of IDH mutation and 1p19q codeletion make no sense. The presence of IDH mutation and 1p19q codeletion, de facto, exclude grade 4 gliomas. Of note, oligoastrocytomas do not exist anymore since the 2016 WHO classification.
- Line 251: words as Fascinatingly have to be avoided in a true scientific manuscript. Line 291: “Considering the limited article space”also sounds inappropriate.
- Figure 5. I cannot trust the analysis. 1p19q codeletion is a well know factor of improved prognosis. 1p19q is pathognomonic of oligodendroglioma which are the gliomas associated to the best prognosis.
- The results exposed in Figures 9, 10 and 11 are few commented. What do the effect of TMP3 on immune infiltration cells brings to the article? These results have to be more detailed, completed and commented or removed.
- The authors tested the effect of TMP3 overexpression in only one cell line. A validation in a second cell line is required. Moreover, these results are not sufficient to conclude about the effects of TMP3. Data regarding the effects of TMP3 depletion using siRNA or shRNA is required.
Reviewer 2 Report
In this article, Huang K. and Xu J. et al performed biomarker exploration from data available in TCGA & GTEx with regards to glioma and its prognosis. Overall, they found that TPM3 & 4 are positively correlated with poorer prognosis of glioma (survival, Cox regression & immune cell infiltration, etc.). In addition, the authors performed in vitro study with U87-MG cells with TPM3 overexpression, showing that TPM3 expression contributes to glioma proliferation, colony forming & migration.
Considering the significance of novel biomarker for Glioma and the depth of analyses the authors presented, I believe the article is of significant quality and interest to the readers of Biology. I suggest the acceptance of this article following minor comments below:
- The authors showed clear correlation between immune infiltration to TPM3 & TPM4 expression. Is there any information whether it is related to altered cytokine /chemokine secretion of the tumor cells?
- Related to above, do the authors observe such notion from the in vitro U87-MG cells overexpression study?
Reviewer 3 Report
In this study, the authors are exploring the biological and clinical relevance of TPM family using various publicly available datasets and resources. Overall, the authors utilize a collection of established methodologies, to give an overview of the TPM-related expression and signaling involved in glioma. However, a deeper integration of the data, together with a more comprehensive comparison with additional external datasets would be needed to validate the clinical implications and prognostic and therapeutic relevance of TPMs and in particular TPM3.
Comments:
1. Line 42-43: an effective biomarker and therapeutic target don’t have to necessarily be the same biological molecule.
2. Text should be proofread to fix minor spelling and Language errors (e.g., line 98: Different Expressed Genes -> Differentially Expressed Genes, line 127: 5x104 -> 5x104)
3. The authors should describe with greater detail the characteristics from the GTEx samples that were used as “normal”. In addition, the authors should further elaborate on the normalization and batch corrections techniques employed to harmonize the TCGA and the GTEx RNA-seq data.
4. The authors should elaborate further on the use of the U87 cell line (a typical glioblastoma model cell line) for the in vitro experiments of the study, although Figure 3J indicated no clinical relationship between TPM3 and Glioblastoma.
5. In addition to gene expression, the authors should explore other multi-omic levels of information (e.g., genomic, proteomic). The integration of such datasets (example: PMID: 33577785) would validate further the findings of the study.
6. Although mentioning the phenotypic effect of TPM3 overexpression, the authors don’t expand on the biological mechanism supporting such effect. For example, is the underlying mechanism of TPM3 involved in cell infiltration (Fig12 E) or in cell proliferation (Fig12 C). Authors should further explore such effects using both gain and loss of function assays of multiple glioma multiple systems to further strengthen their case.
7. The material and methods should be enhanced to include more details regarding the methods that were followed. In particular, regarding the versions (or dates of access) for the tools and databases that were used. Examples:
i. the Xena Browser link is a generic link listing more than 1500 datasets.
ii. ii. the log2FC and adjusted p-value thresholds that were chosen to consider genes as differentially expressed are missing in the DEG analysis using DESeq2.
8. Authors should clarify what the diagnostic value was in the ROC curve of Figure 6.
9. Authors should clarify the version of TIMER they used (1 or 2.0?) and if needed use the latest version of the tool.
Round 2
Reviewer 1 Report
The authors make important efforts to try to improve the quality of the manuscript. There remain some concerns.
1. I am sorry but the quality of the language remains not optimal. For instance, line 14: aim to has to be corrected to aim at. Many other mistakes have to be corrected.
2. Point 4: As all subtypes of gliomas were mixed, it is impossible to determine if TPM expression depends or not on glioma grade. If TPM is more expressed in high grade gliomas than in low grade gliomas, high expression of TPM will be correlated to a worse outcome but will be not an independent prognosis factor. Unfortunately, the authors have not solved the problem and based their methodology on bad-constructed studies. A multivariate analysis is required to conclude.
3. In figure 4, many test were performed and few results would be still significant after the Bonferroni correction which has to be applied.
4. In the response to point 7, authors continue to speak about secondary glioblastoma. This entity does not exist anymore still 2021!
5. Point 9: I still not trust the statistical analysis as codel 1p19q which is a strong prognosis factor has no significance in multivariate analysis
6. Line 336-341: this paragraph corresponds to the description of methods and has no place in the result section
